# Vaccination experiences and decision-making in older adult Korean immigrants living in Canada: A qualitative descriptive study

Ji Yoon Kim[1], Ikyu Park[2], Eunah Cha[3], Giorgia Sulis[4,5], Seungmi Yang[1], Jesse Papenburg[6,7], Patricia Li[6,7], Ananya Banerjee[1]*

1 Department of Epidemiology, Biostatistics and Occupational Health, McGill University, Montreal, Québec, Canada, 2 University of British Columbia, Vancouver, British Columbia, Canada, 3 University of Alberta, Edmonton, Alberta, Canada, 4 University of Ottawa, Ottawa, Ontario, Canada, 5 Ottawa Hospital Research Institute, Ottawa, Ontario, Canada, 6 Department of Pediatrics, McGill University, Montreal, Québec, Canada, 7 Research Institute McGill University Health Centre, Montreal, Québec, Canada

* Ananya.Banerjee@mcgill.ca

## Abstract

Older adult immigrants face unique challenges in accessing healthcare and preventive health measures, including vaccines. While Asian immigrants in North America generally report high vaccine uptake, studies suggest that Korean immigrants may have lower willingness to vaccinate and experience barriers to healthcare utilization. Understanding the vaccination experiences and decision-making processes of this population is critical to addressing disparities in and improving vaccine uptake. Therefore, we conducted a qualitative descriptive study to explore the influenza, pneumococcal, and shingles vaccination experiences, perceptions, and decision-making among older adult Korean immigrants in Canada. Study participants were recruited using convenience, snowball, and purposeful sampling. Semi-structured interviews were conducted with 30 Korean immigrants aged 65 years and older residing in Montreal and Toronto, Canada from September 2023 to July 2024. Interview transcripts and field notes were thematically analyzed, guided by the socio-ecological model. Key themes across intrapersonal-, interpersonal-, institutional-, community-, and policy-levels were identified. Participants reported strong willingness to get vaccinated, largely influenced by healthcare provider recommendations, government guidance, and perceived disease risk. However, gaps in vaccine knowledge, concerns about vaccine safety, and lack of explicit healthcare provider recommendations contributed to non-vaccination. Participants identified trust in the Canadian government and medical professionals as primary motivators to adhere to vaccination guidelines. Some vaccination-specific facilitators and/or barriers were also identified (e.g., financial barriers to shingles vaccination). Strategies to improve vaccine uptake among older adult Korean immigrants could involve supporting healthcare providers or public health efforts to promote vaccinations, encouraging healthcare providers to address

**Data availability statement:** The qualitative data collected for this study includes sensitive personal information that could lead to identification of study participants and staff. Data access requests can be made to the McGill University Institutional Review Board (jonathan.nordland@mcgill.ca).

**Funding:** This work was partly supported by the Tier 2 Canada Research Chair in Communicable Disease Epidemiology held by GS and by a Fonds de recherche du Québec – Santé Senior Clinical Research Scholar award held by PL. The funders had no role in study design, data collection and analysis, decision to publish, or preparation of the manuscript.

**Competing interests:** JP reports grants paid to his institution and unrelated to current work from the Public Health Agency of Canada, the McGill Interdisciplinary Initiative in Infection and Immunity, the Canadian Institutes of Health Research, Merck and MedImmune; and personal fees from Enanta. There are no patents, products in development or marketed products to declare. This does not alter our adherence to PLOS Global Public Health policies on sharing data and materials.

concerns and emphasize safety and benefits of vaccinations using shared decision making, and collaborating with faith-based communities to promote vaccinations.

## Introduction

In Canada, immigrants (i.e., individuals born outside of Canada) represent approximately 23% of the population [1]. Despite constituting a considerable proportion of the Canadian population, research suggests that immigrants, particularly older immigrants, face unique and complex challenges in accessing and receiving vaccinations and that immigrant status is associated with lower vaccination rates [2–8]. This is particularly concerning, as older age is a risk factor for the incidence and severity of various infectious diseases, such as influenza, pneumococcal disease, and shingles [9–12].

Of note, in 2025, adults aged 65 years and older in Canada were recommended by the National Advisory Committee on Immunization (NACI) to receive an influenza vaccine every year and a single pneumococcal vaccine in their lifetime [11,13], and in 2023, adults aged 50 years and older were recommended to receive two doses of the herpes zoster (shingles) vaccine in their lifetime [12]. Influenza and pneumococcal vaccinations are publicly funded for those aged 65 years and older across all provinces and territories in Canada, while shingles vaccination is publicly funded in Ontario, Quebec, Prince Edward Island, and Yukon, among which only Prince Edward Island provides the vaccine for free as per NACI recommendations for adults aged 50 years and older [14]. However, national estimates from 2023 suggest that influenza and pneumococcal vaccination coverage among adults aged 65 years and older were 70.2% and 54.7%, respectively, while only 38.8% of adults aged 50 years and older had received at least one dose of the shingles vaccine [15]. Studies also indicate that vaccination coverage in immigrants is lower by 2.4% for influenza, 10.7% for pneumococcal disease, and 3.8% for shingles, compared to non-immigrants [5,6], presenting a need to address these disparities in vaccination experienced by immigrants.

However, it is important to recognize that immigrants are not a homogeneous group. Different communities have distinct histories, linguistic, cultural, and social norms, as well as pathways to migration that influence health behaviors, including vaccination. Since vaccination decision-making and vaccine hesitancy are vaccine-, context- and culture-specific [16–19], research should account for the heterogeneity among immigrant communities. Understanding the unique experiences, facilitators, and barriers faced by specific immigrant communities is essential to developing tailored and culturally appropriate strategies to improve vaccine uptake.

An older immigrant community that remains heavily overlooked and understudied with respect to vaccination is the Korean immigrant population, one of the largest racial minoritized groups in Canada according to the 2021 Census [20]. This population continues to grow rapidly as well. Between 2001 and 2021, the number of individuals who self-identify as Korean more than doubled, increasing from 101,000–218,000, of which over 60% were born in Korea [20,21].

According to a study conducted using 2003–2009 Canadian Community Health Survey data, Korean adults aged 65 years and older reported lower, although not statistically significant, coverage of influenza vaccination than those of other ethnic groups with the exception of Latin American, West Asian/Arab, and Black older adults [22]. This was the only study in the Canadian context that reported vaccination coverage among Korean older adults, but was not specific to Korean immigrant older adults. Similarly, a study conducted using 2009–2012 REACH US Risk Factor Survey reported that Korean American adults aged 50 years and older reported lower influenza vaccination coverage than those of other ethnic groups with the exception of Non-Hispanic Black adults, and that Korean adults aged 65 years and older reported lowest coverage of pneumococcal vaccination among population groups analyzed [23]. Other studies on vaccine willingness and healthcare service use among Korean adults in North America, primarily conducted in USA and focusing on COVID-19 vaccinations, suggest that Korean adults are less willing compared to other Asian adults to get vaccinated and less likely to access and utilize healthcare services, and that various barriers negatively impact Korean immigrants' health and healthcare utilization, such as lack of social support, language difficulties, geographic constraints, and economic challenges [19,24–26].

While data on the burden of vaccine-preventable diseases on Korean older adult immigrants are not available, given that NACI recommends older adults receive influenza, pneumococcal, and shingles vaccines [11–13], but that Korean immigrants have lower vaccination willingness and healthcare service use [19,24–26], there is a pressing need to better understand the decision-making processes of Korean immigrants, to inform tailored strategies to improve vaccine uptake and hesitancy. Qualitative studies are particularly well suited to address the "how" and "why" of vaccination behaviours and decision-making processes within specific contexts and immigrant communities as it enables an in-depth exploration of these complex processes [27,28].

To address these gaps, we conducted a qualitative descriptive study to gain an in-depth understanding of influenza, pneumococcal, and shingles vaccination experiences and decision-making among Korean immigrants aged 65 years and older in Canada. The specific objectives of this study were to: 1) explore vaccination experiences of older adult Korean immigrants in Canada, including their attitudes, perceptions, and perspectives on vaccination, focusing on influenza, pneumococcal, and shingles vaccination; and 2) identify factors that facilitate or hinder uptake of influenza, pneumococcal, and shingles vaccines among older adult Korean immigrants in Canada. Findings of our study will inform evidence-based, culturally appropriate, and targeted strategies to improve vaccine uptake in this understudied population. Furthermore, while this study focuses on older adult Korean immigrants in Canada, the findings have broader relevance for global public health. In 2022, 43,000 Korean citizens emigrated to OECD countries such as the United States, Canada, and Japan [29]. The considerable size of Korean diaspora around the world highlights the need for increased efforts to better understand the health of Korean immigrants [30]. While the findings from this study are specific to Canada, they may be applied to other countries with similar vaccination policies and cultural contexts.

## Methods

We report study methods and findings in accordance with the standards for reporting qualitative research (SRQR) guidelines.

### Ethical statement

This study was approved by the McGill University Institutional Review Board (A06-B28-23B (23-04-093)). We obtained written and verbal informed consent to participate in interviews and provide self-reported demographic and vaccination data prior to any data collection. Participants were informed at each point of contact that study participation was completely voluntary, and that they could withdraw from the study at any point. Study participants received a $50 gift card in appreciation for their time. All study data were kept strictly confidential, accessible only to the immediate research team (JK, IP, PL, AB), and stored in a secure internal database.

## Study design and setting

We conducted a qualitative descriptive study, a methodological approach well-suited for capturing participants' perceptions, perspectives, and experiences with a health-related phenomenon to identify problems and inform practical changes, particularly in understudied research areas/populations [27,28,31].

The study was conducted from September 2023 to July 2024 in Montreal and Toronto, home to approximately 40% of Canada's Korean immigrant population [21]. Study participants were recruited from September 1, 2023 to June 30, 2024. Data collection included self-administered questionnaires and semi-structured interviews.

## Research team

Our research team included researchers with experience and expertise in vaccination research (JK, GS, JP, AB), health equity research and qualitative methodologies (JK, IP, EC, GS, SY, PL, and AB) as well as clinician researchers (JP and PL). JK, IP, EC, and SY are self-identified Korean immigrants. AB, GS and PL are self-identified immigrants.

## Recruitment and data collection

We recruited study participants using convenience, snowball, and purposeful sampling methods [32,33], as multi-model strategies are recommended to ensure that a diverse study population that meets the inclusion criteria is recruited [34]. A recruitment poster was distributed at various Korean community organizations, churches, and grocery stores in Montreal and Toronto. All participants were also encouraged to promote study participation to their personal connections and within their networks. To elicit diverse perspectives, we purposefully sought a study sample composed of self-identified men and women as well as individuals with varying time since immigration and English fluency as such factors are important social determinants of vaccination that intersect to differentially shape vaccination experiences and decision-making [7,8,35,36].

Inclusion criteria for study participation were: 1) 65 years of age or older; 2) born in Korea and self-identifying as Korean; 3) able to read, write, and speak English or Korean; 4) eligible to receive publicly-funded vaccinations; and 5) currently residing in the Montreal Metropolitan Area or the Greater Toronto Area. Exclusion criteria were: 1) unable to travel to the interview site or use videoconference (Zoom); and 2) unable to provide informed consent.

The interview guide focused on the following topic areas and included probes for influenza, pneumococcal, and shingles vaccinations: 1) experiences in accessing and utilizing vaccinations and healthcare services in Canada and Korea; 2) vaccination perceptions; and 3) vaccination decision-making process. Prior to conducting the interview, we collected participants' demographic and self-reported vaccination data through self-administered questionnaire. The final sample size was determined by the research team through continuous reflection and discussion about the diversity of participants and was informed by data saturation [37].

## Data analysis

Interviews were audio-recorded and transcribed using artificial intelligence software (ClovaNote). Interviews conducted in Korean were translated to English using artificial intelligence software (Papago). All transcriptions and translations were independently reviewed and verified for accuracy by the first author prior to uploading the data into NVivo 14 for data coding and analysis.

We conducted an inductive and deductive thematic analysis to identify common themes and sub-themes across the data [38,39]. The first few transcripts were read independently by the first author to extract codes and develop the initial coding list, which was verified by the second author (IP) and reviewed by PL and AB. A codebook was developed in NVivo and refined iteratively throughout the analyses. All interview transcripts were independently coded by the first author and reviewed and verified by the second author to ensure agreement and consistency between coders [40]. Any discrepancies were resolved through discussion by the two authors.

Upon completing coding of all transcripts, the first and second authors created a summary sheet of themes and sub-themes identified, as well as illustrative quotes from the transcripts. This summary was reviewed with PL and AB to generate final themes and sub-themes.

We organized emerging themes and sub-themes using the socio-ecological model (SEM). Introduced by Bronfenbrenner, the framework demonstrates that personal and environmental factors interact and intersect to influence one's health and health behaviours [41,42]. We chose to present our results using the SEM over more vaccination-specific frameworks such as the World Health Organization's behavioural and social drivers of vaccination [16], which was conceptualized and developed using the SEM [43], to highlight the interaction of factors across different layers in shaping vaccination decision-making, as well as the importance of higher-level, more structural and systemic factors that impact vaccination decision-making, partially through downstream effects on individual-level determinants of vaccination. Notably, the domains in the BeSD of vaccination are related to and reflect various SEM levels [44]. For instance, the thinking and feeling domain corresponds to the intrapersonal level, whereas the social processes domain contains themes that belong to the interpersonal, institutional, community, and policy levels and the practical issues domain themes reflect intrapersonal, interpersonal, institutional, and policy-level factors.

While multiple variations of the framework exist, we employed the 5-layer version to highlight the complex interaction of individual- (i.e., individual characteristics), interpersonal- (i.e., formal and informal social networks and support systems), institutional- (i.e., social institutions with organizational characteristics and rules that govern the institutions), community- (i.e., relationships between organizations, institutions, and informal networks), and policy-level (i.e., laws and policies) factors in shaping one's vaccination decision-making [45]. This well-established framework has been used extensively in previous research on childhood immunizations, adolescent vaccinations, adult vaccinations, and COVID-19 vaccinations [46–53].

Several approaches were employed to achieve rigor and quality, to ensure that the study findings are credible, dependable, confirmable, and transferable [54,55]. Interviews were audio-recorded, transcribed, translated, and independently analyzed by two researchers for credibility and dependability. All transcripts were checked against the original recording to ensure accuracy. Peer debriefing was conducted regularly for credibility. Upon identification of final themes and sub-themes, study participants were followed up a maximum of three times for member checking for credibility – to ensure that our study findings reflected the participants' experiences and to receive additional feedback. Field notes with thick description were also taken throughout the research process for transferability and to maintain an audit trail for credibility, dependability, and confirmability.

## Results

A total of 30 Korean immigrants aged 65 years and older residing in Montreal or Toronto participated in semi-structured interviews, at which saturation was reached (Table 1). One interview was conducted via videoconference and all others were conducted in-person. The interviews lasted between 40 and 110 minutes (mean of 61.5 minutes). Most participants resided in the Toronto area, were married or living with a partner, identified as Christian, and had completed college or university-level education. Almost all study participants reported having a family physician. At the time of the interview, 10 participants (33.3%) reported having received all three recommended vaccines, 9 (30.0%) reported having received two of three vaccines, 7 (23.3%) reported having received one of three vaccines, and 4 (13.3%) reported having received none of the vaccines. All interviews were conducted in Korean.

Fig 1 presents the results of this study.

### Intrapersonal-level

**1. Perceived disease risk.** 1.1. *Perceived disease risk to self*: All participants reported that their perception of how much risk they are at of acquiring and/or suffering from disease plays a key role in their vaccination decisions.

**Table 1. Sociodemographic characteristics and vaccination statuses of study participants.**

| Characteristics | Number of participants (%) |
|---|---|
| Age group | |
| 65–69 years | 9 (30.0) |
| 70–74 years | 7 (23.3) |
| 75–79 years | 10 (33.3) |
| 80 years and older | <5 |
| Sex at birth | |
| Male | 18 (60.0) |
| Female | 12 (40.0) |
| Immigration status | |
| Obtained Canadian citizenship | 27 (90.0) |
| Permanent resident | <5 |
| Area of residence | |
| Montreal Metropolitan Area | <5 |
| Greater Toronto Area | 28 (93.3) |
| Years lived in Montreal/Toronto | |
| 10 years or less | <5 |
| 11–20 years | 5 (16.7) |
| 21–30 years | 8 (26.7) |
| 31–40 years | <5 |
| 41–50 years | 7 (23.3) |
| 51 years or more | <5 |
| Current marital status | |
| Married/living with a partner | 23 (76.7) |
| Divorced | <5 |
| Widowed | 5 (16.7) |
| Prefer not to answer | <5 |
| Religion | |
| Christianity | 27 (90.0) |
| Buddhism | <5 |
| No religious affiliation | <5 |
| Highest level of education | |
| Secondary school graduation | <5 |
| Did not finish college/university | 5 (16.7) |
| College/university graduation | 16 (53.3) |
| Post-college/university education | <5 |
| Prefer not to say | <5 |
| Total annual household income | |
| Less than $20,000 | <5 |
| $20,000 or more, but less than $50,000 | 12 (40.0) |
| $50,000 or more, but less than $100,000 | 9 (30.0) |
| $100,000 or more | <5 |
| Don't know/prefer not to say | 6 (20.0) |
| Has family physician | |
| Yes | 29 (96.7) |
| No | <5 |

*(Continued)*

**Table 1.** (Continued)

| Characteristics | Number of participants (%) |
| --- | --- |
| Influenza vaccination in past 12 months | |
| Yes | 22 (73.3) |
| No | 8 (26.7) |
| Pneumococcal vaccination ever | |
| Yes | 14 (46.7) |
| No | 10 (33.3) |
| Don't know | 6 (20.0) |
| Shingles vaccination ever | |
| Yes | 19 (63.3) |
| No | 10 (33.3) |
| Don't know | <5 |

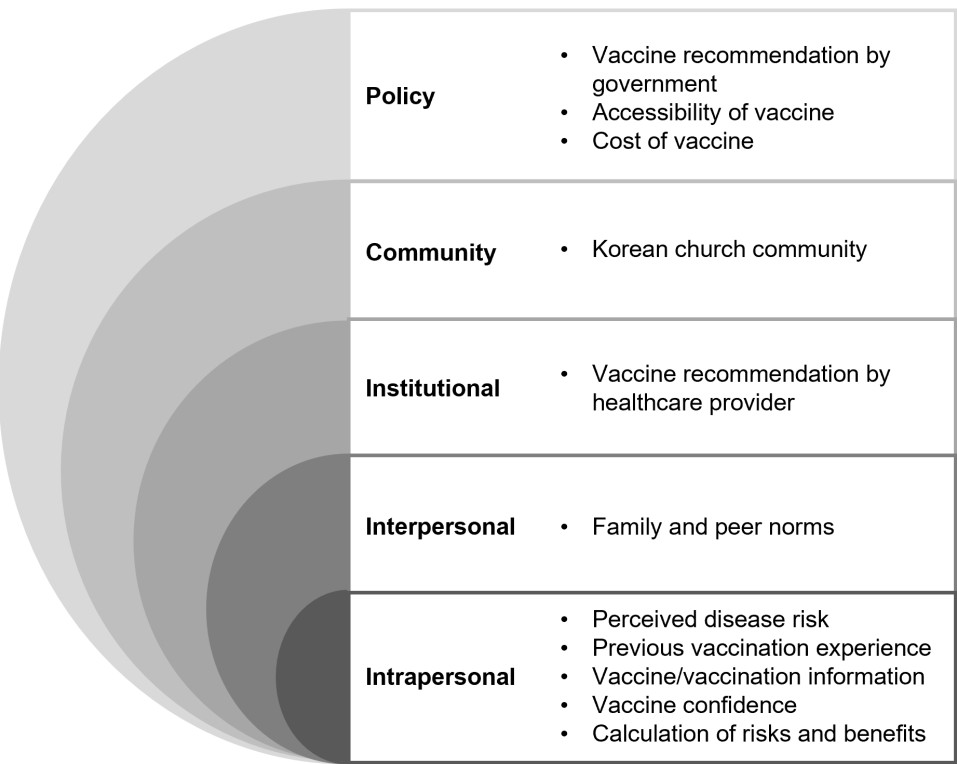

**Fig 1. Summary of themes and sub-themes by SEM category.** Intrapersonal factors represent individual characteristics (e.g., knowledge, attitudes, behaviours, etc.). Interpersonal factors include social networks and support systems. Institutional factors refer to social institutions with organizational characteristics and rules that govern the social institutions, as well as interactions with such social institutions. Community factors encompass relationships between organizations, institutions, and informal networks. Public policy factors include laws and policies.

Many expressed feeling less healthy as they age and noted their increased susceptibility to acquiring infections and/or developing severe complications. This heightened awareness of vulnerability due to aging-related health decline served as a strong motivator for vaccination.

*"Well, because I'm old now, I don't really have that immunity. So I'm getting it out of precaution because I don't want to get sick. […] But in our case, for older people like us, if we catch a cold then it lasts a long time. We don't get better easily either. And even if we think we got better, side effects last like 2-3 weeks, or even over a month. So I just get vaccinated no matter what."* (Participant #20)

Conversely, some participants expressed that if a disease or its severe complications were uncommon within their social circles, it seemed less relevant or important to them, leading to their decision not to get vaccinated.

*"But there weren't that many people that had shingles around me. That's probably why I didn't get the vaccine. Like, with the cold, so many people get it. So I want to avoid that, but with shingles, there's very few people that had it around me. So I didn't get it. But if there were like 3-4 people that I know that had shingles, then I should go get my vaccine right away. But it wasn't that important, like there weren't that many people that had it, so I didn't."* (Participant #13)

1.2. *Perceived disease risk to others*: Participants' vaccination decisions were also influenced by concerns about how their illness could impact others. For instance, participants reported getting vaccinated to remain healthy and prevent burdening their family members.

*"…like if I fall ill, then I'm making everyone worried right? Especially my children, and I'd be a big burden on my wife, if I fall ill. For example, if I catch a cold and get sick for like ten days or two weeks or something like that, then the people around me have a hard time. So like because I'm getting weaker, it's more important for me to not be sick. To be healthy."* (Participant #14)

Beyond personal health, many participants also expressed a strong sense of social responsibility to become vaccinated to prevent spreading the disease to others. Some even framed non-vaccination as an ethical concern, describing it as *"committing a sin" (Participant #8)* by potentially spreading illness to others.

*"… if I catch a cold, then that affects my family. My husband also has side effects from stroke, so he's frail, and if I go out and about to socialize, it might spread to him too. And I go out in the public a lot, and I might spread it. So out of responsibility at least, I should get vaccinated."* (Participant #8)

**2. Previous vaccination experience.** Most participants recalled having no or mild side effects from previous vaccinations and reported that these experiences did not influence their vaccination decisions as *"it doesn't make sense to not get vaccinated because it hurts a little" (Participant #6)*. However, those that had experienced serious adverse effects in the past expressed increased hesitancy about further vaccination.

*"After we had such a hard time back then, I did think, 'Oh, I got vaccinated and am now suffering from bad illness, am so seriously ill and in such severe pain', so I considered not getting vaccinated when I was lying down at home. I thought about that, but if everyone around me didn't get vaccinated, then maybe, but I just thought that it was maybe because I wasn't too healthy back then, so I just got vaccinated again."* (Participant #29)

**3. Vaccine/vaccination information.** 3.1. *Knowledge about vaccines*: Many participants reported that being informed about vaccines (i.e., knowing what the vaccine is for, how the vaccine works, their eligibility for vaccination, etc.) facilitates their decision to become vaccinated.

*"Well, with shingles, for example, when they recommended that I should get the vaccine for that, they said that you might get a mild version or a severe version of shingles, that you might get it when you're young or when you're old,*

*that you can get it any time, so it's good to get vaccinated. So because they explained all of this, that's how I made the decision." (Participant #26)*

However, a few participants were unaware of the existence of certain vaccines, particularly the pneumococcal vaccine, which contributed to their non-vaccination. These participants reported that *"no one really talked to [them] about it or recommended it" (Participant #21)* and asked the interviewer questions about the vaccine during the interview.

3.2. *Knowledge about where to get vaccinated*: Most participants reported knowing at least one location where they could receive vaccinations. However, some were unsure where to access pneumococcal and/or shingles vaccines, which contributed to their non-vaccination. Participants also reported not knowing where they can seek information about vaccinations, and that even when they actively sought information (e.g., by going to the pharmacy), they were unable to receive it.

*"Sometimes I want to get a vaccine and ask around, but there are no institutions or people that can answer me that well. […] I think I had shingles last time… Right, shingles. I had that a few months ago. So someone told me that if I go to the [local community health centre] or something they would take care of it. So I asked around but people told me that they don't give those vaccines. So I didn't really get a chance to talk more about it. […] I went to the pharmacy and I bought the medicine and asked, and they told me that they don't know. And I haven't asked a doctor yet. So the pharmacy should know, […] but they said that they don't know and I let it go, but it would be nice to get that." (Participant #1)*

**4. Vaccine confidence.** 4.1. *Perceived benefits and effectiveness of vaccination*: Almost all study participants expressed confidence in the benefits of vaccination (i.e., good and important) but also voiced concerns about safety and necessity in certain cases. Nearly all participants believed that vaccinations are beneficial for disease prevention. However, some questioned whether vaccination was necessary for individuals in good health.

*"It's good to get vaccinated. Because it's for prevention. So the intention is good, but… I don't know/ I wonder if healthy people really need to get vaccinated. Prevention is good, but if there's nothing wrong with you, if you're perfectly healthy, there's no reason to…" (Participant #21)*

4.2. *Perceived safety of vaccine*: Concerns about vaccine safety were among the most commonly cited reasons for non-vaccination. Some participants expressed uncertainty about vaccine ingredients and potential risks.

*"We don't know the things that are in vaccines, the bad stuff or whatever. How would we know? The ingredients in vaccines, and what harmful things there are to getting vaccinated, I don't know. Only the doctors know. But for doctors, it's better for them to have more people come and get vaccinated, right?" (Participant #20)*

4.3. *Trust in vaccine*: During the interviews, many participants described their *"baseline faith in vaccines" (Participant #24)*, believing that vaccination is generally beneficial and should be trusted.

*"So I thought that rather than not getting vaccinated, it must be better to get vaccinated. So I can trust it like that, and I have faith in it." (Participant #19)*

**5. Calculation of risks and benefits of vaccination.** Some participants expressed that they evaluate and balance the risks and benefits of vaccination, in particular the likelihood of suffering from vaccine side effects against the probability of the vaccine-preventable disease, before making a decision regarding vaccination.

*"I think that it's all about the probability. […] And technically, risk is this… uncertainty. The fact that you could get better or worse from it, it's all a risk. With investing in stocks, you also have to take the risk. You might earn a lot, but also*

*lose a lot. [...] But with vaccines, like, who would want to get vaccinated? There are risks associated with vaccines too. There are risks of side-effects." (Participant #9)*

### Interpersonal-level

**6. Family and peer norms.** Some participants identified family and friends' recommendations as important influences on their vaccination decisions. Generally, participants reported that when their family and friends received a vaccine, it encouraged them to participate as well.

*"If all my friends tell me that they've been vaccinated, then I need to as well." (Participant #14)*

Many participants reported that most of their friends attend church and that their social circles and church community overlap, thus likely reinforcing shared health behaviours and norms.

*"The church is [a big part of my life]. Like, we do other stuff like exercising, playing golf and table tennis and stuff. But my friends all go to church. Like… most immigrants do, no? Almost all… of my friends are like that." (Participant #14)*

### Institutional-level

**7. Vaccine recommendation by healthcare provider.** Most participants identified healthcare provider recommendation as the most influential factor in their vaccination decision-making process and cited physicians' expertise as the main reason for their trust in these recommendations.

*"I would say that the recommendation of my family doctor would be the most important [reason for vaccination], because he's the one who knows best and he's the expert." (Participant #5)*

*"If the doctor told me to, then I would have [gotten vaccinated]. But I've never been recommended it, and I thought that I didn't really need the vaccine yet." (Participant #19)*

However, many participants noted that their healthcare providers did not spend much time discussing vaccines with them or ensured that they understood their purpose. While many expressed a desire to become more informed, participants reported that they generally followed vaccination recommendations, regardless of the amount of time spent with healthcare provider to discuss vaccinations.

*"He always recommended them to us, so I went along with his recommendations whenever he made them. [...] Well... first of all, the problem was that we didn't fully understand him. I also don't think that the doctor explained to us much in detail. He told us what we need it for, and we just followed his recommendation. That's what happened." (Participant #30)*

### Community-level

**8. Korean church community.** Many participants identified the church community to be an important part of their lives and played a key role in their vaccination decision-making. They reported that their church community serves as their primary source of vaccine information. Participants also noted that pastors and congregation members champion and encourage vaccination among their church community.

*"And at the church too, in the case of our church, when the government has such policies [vaccine recommendations], then we communicate those through the pastor's column or whatever so that people can benefit. [...] [I]f there's any members that don't have access to newspapers or whatever, then we contact them to let them know, so yeah. Our church is a very small group-centered church to begin with, so our leaders take care of things like that very well. So we all share a similar mindset, so if we need to get vaccinated, there's a lot of peer pressure to do so." (Participant #25)*

**Policy-level**

**9. Vaccine recommendation by government.** Many participants cited government recommendations as an important reason for vaccination. Most identified media, such as *"mass media, newspapers, and so on" (Participant #23)* as their source of information about government vaccination policies and guidelines.

*"I'm just going with the government rules. Following the government rules. The government wants all of us regular people to get vaccinated, right? Because it's a loss for the government if someone doesn't get vaccinated and gets sick. So like, I trust the government. I trust the government and follow their lead." (Participant #16)*

During the interviews, many participants cited their trust in the government in ensuring standard safety and quality of the vaccines for public health protection as primary reason for following government recommendations for vaccinations.

*"You have to trust. You have to trust because if there's some vaccine center, then those vaccine centers will be audited by the Ministry of Health or whatever. They will make sure that while distributing, the distribution trucks and fridges and stuff are all fine, in case the vaccines go wrong... It makes no sense that they can't do that. So I have no doubts about that." (Participant #23)*

The high level of trust, particularly in the Canadian government, among older adult Korean immigrant community was found to be shaped by participants' past experiences in Korea, where corruption, conflict, and poverty influenced their perceptions of authority. Participants also reported feeling gratitude towards the Canadian government when comparing their past experiences with the government in Korea to their current experiences with the government in Canada.

*"I immigrated here when things were rough in Korea… Well, the 6.25 incident (the Korean war) occurred in my first year of elementary school, so we lived in that corrupt, the Korean society after the war, so I didn't have any rejections when I got here where everything is in place. So like… I should appreciate the government." (Participant #29)*

**10. Accessibility of vaccine.** All study participants who knew where they could receive vaccines reported that they found vaccinations to be easily accessible and convenient to receive.

*"I just go when they tell me to come… There wasn't much for me that was inconvenient." (Participant #14)*

While many participants expressed their preference to receive vaccinations from their family physicians as *"it's better to get vaccinated by professionals"* and they *"have more trust in people with more expertise" (Participant #28)*, they reported having received most vaccines at pharmacies due to convenience.

*"I like it when the pharmacist gives it to me. Like… well, I don't necessarily like it, I'd rather have the family doctor give it to me. And like she would, if I asked her to. But I think that, with the time and everything, there's so many people right? So it's more convenient for me to get it from my pharmacist. […] It's easier. More convenient." (Participant #14)*

**11. Cost of vaccine.** Cost was generally a factor that facilitated vaccine uptake among participants, who reported that no cost motivated them to get vaccinated. However, some participants cited cost as a barrier to vaccination, particularly for the shingles vaccine.

*"I have heard that I have to pay for the shingle shot if I am older than 75. So, I felt like the country does not protect people over 75 years old, so I decided to not get a shingle shot, and also, I did not want to pay for it." (Participant #17)*

## Discussion

This qualitative descriptive study explored the influenza, pneumococcal and shingles vaccination experiences and decision-making processes of older adult Korean immigrants in Canada. Consistent with established frameworks of vaccination [16,17], our findings indicate that vaccination decisions are driven by the complex interplay of various intrapersonal-, interpersonal-, institutional-, community-, and policy-level factors. Within the different SEM layers, the most commonly reported motivators of vaccination were perceived disease risk to themselves and others (intrapersonal), family and peer norms (interpersonal), healthcare provider recommendation (institutional), church support (community), and government recommendations (policy), among which perceived disease risk to themselves and others was the most commonly reported motivator while healthcare provider recommendation was cited as the most important. Conversely, concerns about vaccine safety was the most common and important reason for not getting vaccinated, consistent with findings previous literature on factors influencing vaccination and vaccine hesitancy among immigrant and migrant populations [7,8,35,36].

Our finding that healthcare provider recommendation is one of the most important determinants of vaccination in older adult Korean immigrants is consistent with previous research on vaccination decision-making among older adult and immigrant populations [2,56–60]. During interviews, participants also reported a desire to spend more time with and receive sufficient information to guide vaccination decision-making from healthcare providers who lacked the time and resources to educate them. Previous research suggests that immigrants often lack vaccine information and awareness [8,35,61], and that healthcare providers are immigrants' primary source of vaccination information [57,58]. As such, immigrants' limited knowledge and awareness about vaccines, combined with the central role of healthcare providers as sources of vaccination information, may explain the participants' specific desire to spend more time with their healthcare providers to receive guidance on vaccination.

Government recommendation was also a commonly cited reason for vaccination among participants. Many reported high levels of trust in the Canadian government and expressed their desire to adhere to government recommendations on vaccination. Participants compared the poverty, government corruption, and conflict experienced during their time in Korea to their current experiences with the government in Canada and reported that such experiences from Korea have fostered their trust in and gratitude towards the Canadian government. While gratitude is a common emotion reported by immigrants [62–64], it is possible that due to the historical and political contexts leading up to the participants' immigration to Canada, this emotion is particularly relevant to Korean immigrants aged 65 years and older. The participants' feelings of gratitude and indebtedness likely contributed to their trust in the Canadian government and desire to adhere government vaccination recommendations.

We found that religion and religious service attendance play important roles in vaccination decision-making processes of older adult Korean immigrants. Study participants reported that their social circles mostly existed in their church communities, which not only served as their primary source of vaccine information, but also actively recommended and motivated participants to get vaccinated. During interviews, many participants described having "faith" in vaccines, and that they would be "committing a sin" by not getting vaccinated and potentially acquiring a contagious disease that they would then spread to others. Our findings are consistent with previous research that suggests that religiosity and religious

service attendance are associated with improved health outcomes [65–67], as well as preventative healthcare use and behavior [68,69]. Faith-based organizations can play a critical role in effective dissemination of public health and vaccination campaigns, as demonstrated during the COVID-19 pandemic [70], to addressing vaccine hesitancy, vaccine uptake, as well as vaccination disparities in marginalized and underserved populations [70–72]. While estimates of religious denominations are not available among immigrants from Korea in Canada, data from the U.S. suggest that approximately 59% of Korean Americans self-identify as Christian [73]. Therefore, our findings suggest that religion can be an important leveraging factor that can be taken advantage of to improve vaccination coverage in older adult Korean immigrants.

In our study, participants were generally very willing to get vaccinated and expressed a sense of obligation to do so. However, despite their high vaccination willingness, vaccination coverage rates were not very high, and many conveyed conflicting thoughts and feelings about vaccine effectiveness and safety. For instance, all participants agreed that vaccines help prevent disease or mitigate symptom severity. However, many also questioned whether vaccines are truly effective and safe. This is consistent with previous research showing that while vaccine hesitancy and willingness may be closely linked to uptake, they are not equivalent to uptake [16,74].

Of the three vaccinations examined in our study, the vaccination rate among participants was highest for influenza, followed by shingles and pneumococcal disease. This is in contrast to the results from a recent national vaccination coverage survey among the general adult population in Canada, which indicated that among older adults, vaccination rates are highest for influenza, followed by pneumococcal disease and shingles [15]. During our interviews, many participants mentioned knowing or having heard of people who had suffered from shingles, which motivated them to seek vaccination. High levels of awareness about and uptake of the shingles vaccine observed among participants may be reflective of what Korea has experienced in recent years – the high burden of herpes zoster infection, an increased awareness about the disease and vaccine following the 2022 presidential election in Korea during which public funding of the vaccine was used as an election strategy, subsequent efforts of multiple regional jurisdictions to provide free shingles vaccination, and competitive shingles vaccine marketing strategies employed by pharmaceutical companies [75–77]. Research on health practices and behaviors of immigrants suggests that immigrants maintain close ties with their places of origin and that their health-related behaviors and decisions are transnational [25,78–80]. The transnational ties of study participants may have contributed to their high awareness of shingles disease and vaccine. This highlights the importance of accounting for immigrants' connections to their countries of origin and therefore the epidemiology of diseases and vaccine uptake, awareness, and ongoing and previous vaccine promotion efforts of their countries of origin when designing interventions to increase vaccination coverage.

The findings from our study suggest that repeated vaccination recommendations from healthcare providers may facilitate vaccination among older adult Korean immigrants. The study results also demonstrate the need to support healthcare providers in dedicating more time to vaccine discussions, particularly those that work closely with immigrant communities and are able to provide language-concordant care, to improve vaccination rates among older adult Korean immigrants. However, such one-on-one interventions may not be as feasible given the time and resource constraints. An alternative approach that circumvents this restriction could involve group educational sessions for older adult Korean immigrants in the Korean language and in Korean community settings (i.e., churches, Korean schools, Korean community associations, etc.), led by healthcare providers and in collaboration with religious leaders, as educational interventions for communities have been demonstrated to effectively improve vaccine hesitancy and uptake [56,81–83]. Furthermore, since the level of trust in government is very high in this population, our findings suggest that government-led public health campaigns could play a role in addressing knowledge gaps and vaccine safety concerns, ultimately helping to reduce hesitancy and increase uptake.

Our study is not without limitations. First, our sampling strategies were not random, which may have over-represented individuals with Canadian citizenship, higher education, and Christian faith compared to the overall older adult Korean immigrant population in Canada, potentially introducing bias into our findings. Second, we excluded individuals unable

to pay for transportation or participate via videoconference, which may have biased the results. Third, we only recruited study participants from Toronto and Montreal. While approximately 40% of the Korean immigrant population in Canada reside in the two areas and Toronto is where the highest number of Korean immigrants reside in Canada [21], our findings may not be generalizable to older adult Korean immigrants living in other parts of Canada. Fourth, our study focused on understanding only influenza, pneumococcal, and shingles vaccination experiences and decision-making. Given the vaccine-specific nature of hesitancy and uptake [16,18], our findings may not apply to other recommended vaccines for older adults in Canada. Lastly, the study population was purposefully restricted to older adult Korean immigrants living in Canada to identify culturally-relevant and context-specific factors that drive vaccination decision-making processes in this population. While some findings may be relevant to other immigrant communities in Canada or Korean immigrants living in other countries with similar vaccination schedules, policies, and cultures, the restriction of our study population may limit the transportability of our findings.

## Conclusions

In conclusion, we identified intrapersonal-, interpersonal-, institutional-, community-, and policy-level factors influencing influenza, pneumococcal, and shingles vaccination decision-making in older adult Korean immigrants in Canada. Overall, most participants were willing to get vaccinated, although hesitant about vaccines. Strategies to improve uptake in this population could involve reminding healthcare providers to give explicit and repeated recommendations for vaccinations, supporting healthcare providers or public health efforts to hold group educational interventions about vaccinations and provide vaccine information and knowledge, and working with faith-based communities to promote vaccinations. Our findings also highlight the importance of historical contexts of countries of origin and transnational ties in shaping immigrants' vaccination decision-making and health decisions. Further studies among Korean immigrants of different age groups and in cities beyond Montreal and Toronto on their vaccination decision-making processes, reasons for non-vaccination, and potential strategies to improve vaccination coverage are needed for recommended vaccinations with low uptake rates as vaccination decision-making is context- and vaccine-specific.

## Supporting information

**S1 Table. SRQR Checklist.**
(DOCX)

## Acknowledgments

We thank all study participants who shared their experiences during interviews. We would also like to thank the support of Korean community organizations, grocery stores, and churches in Toronto and Montreal. PL and AB are co-senior authors of this manuscript.

## Author contributions

**Conceptualization:** Ji Yoon Kim, Ananya Banerjee.

**Data curation:** Ji Yoon Kim.

**Formal analysis:** Ji Yoon Kim, Ikyu Park, Eunah Cha, Giorgia Sulis, Seungmi Yang, Jesse Papenburg, Patricia Li, Ananya Banerjee.

**Methodology:** Ji Yoon Kim, Patricia Li, Ananya Banerjee.

**Project administration:** Ji Yoon Kim.

**Supervision:** Patricia Li, Ananya Banerjee.

**Writing – original draft:** Ji Yoon Kim.

**Writing – review & editing:** Ji Yoon Kim, Ikyu Park, Eunah Cha, Giorgia Sulis, Seungmi Yang, Jesse Papenburg, Patricia Li, Ananya Banerjee.

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
