## [Decision Letter · Decision Letter 0]

26 Oct 2025

PGPH-D-25-01841

Vaccination experiences and decision-making in older adult Korean immigrants living in Canada: A qualitative descriptive study

Dear Dr. Ji Yoon Kim,

Thank you for submitting your manuscript to PLOS Global Public Health. After careful consideration, we feel that it has merit but does not fully meet PLOS Global Public Health’s publication criteria as it currently stands. Therefore, we invite you to submit a revised version of the manuscript that addresses the points raised during the review process.

We look forward to receiving your revised manuscript.

Kind regards,

Baldeep Kaur Dhaliwal, PhD

Academic Editor

Journal Requirements:

i. State the initials, alongside each funding source, of each author to receive each grant. For example: "This work was supported by the National Institutes of Health (####### to AM; ###### to CJ) and the National Science Foundation (###### to AM)."

2. Please ensure that your Ethics Statement is available in its entirety at the beginning of your Methods section, under a subheading 'Ethics Statement'.

3. Please provide separate figure files in .tif or .eps format.

4. We have noticed that you have uploaded Supporting Information files, but you have not included a list of legends. Please add a full list of legends for your Supporting Information files after the references list.

5. For studies involving third-party data, we encourage authors to share any data specific to their analyses that they can legally distribute. PLOS recognizes, however, that authors may be using third-party data they do not have the rights to share. When third-party data cannot be publicly shared, authors must provide all information necessary for interested researchers to apply to gain access to the data. (https://journals.plos.org/plosone/s/data-availability#loc-acceptable-data-access-restrictions)

Reviewers' comments:

Reviewer's Responses to Questions

**Comments to the Author**

1. Does this manuscript meet PLOS Global Public Health’s publication criteria?

Reviewer #1: Yes

Reviewer #2: Yes

2. Has the statistical analysis been performed appropriately and rigorously?

Reviewer #1: Yes

Reviewer #2: N/A

3. Have the authors made all data underlying the findings in their manuscript fully available (please refer to the Data Availability Statement at the start of the manuscript PDF file)?

Reviewer #1: Yes

Reviewer #2: Yes

4. Is the manuscript presented in an intelligible fashion and written in standard English?

Reviewer #1: Yes

Reviewer #2: Yes

Reviewer #1: Congratulations on your manuscript addressing an important and timely topic, as vaccine confidence and the reduction of inequities among migrant populations remain key global public health priorities. You have undertaken a rigorous and well-reported study. Your paper would be further strengthened by more clearly situating the findings within the existing body of literature and by providing more specific recommendations to guide future research.

• Intro (p.4): “…shingles vaccination is publicly funded in 4 public health jurisdictions, among which only one provides the vaccine for free as per NACI recommendations for adults aged 50 years and older” > I would suggest specifying the four jurisdictions and indicating which province offers the shingles vaccine for free.

• Intro (p.4-5): “Studies also indicate that vaccination coverage in immigrants is lower by 2.4% for influenza, 10.7% for pneumococcal disease, and 3.8% for shingles, compared to non-immigrants…” > Including data on vaccination coverage among older Korean immigrants, if available, would strengthen this section.

• Intro (p.5-6): “Studies on vaccine willingness and healthcare service use among Korean immigrants conducted in North America suggest that Korean immigrants…” > It would be important to situate your study within the existing evidence base and to clearly identify the specific gaps your research addresses. Are the relevant studies based in Canada or primarily in the U.S.? Do they focus on child vaccination or on older sub-populations? Are they centred on vaccine uptake specifically, or do they address broader patterns of healthcare service use?

• Intro (p.6): “…there is a pressing need to better understand the decision-making processes of Korean immigrants, to inform tailored strategies to improve vaccine uptake and hesitancy.” > Is there evidence indicating not only lower vaccination coverage rates but also a higher risk of influenza, shingles, or pneumococcal disease among older Korean immigrants, which would further underscore the urgency of the issue?

• Methods: I would suggest introducing Bronfenbrenner’s socio-ecological model (SEM) in the methods section, rather than at the beginning of the results section, along with a brief presentation of the SEM categories.

• Results: p. 14: “Institutional factors refer to organizational characteristics and rules that govern social institutions.” > In the results section, it might be helpful to clarify that institutional factors include both health facilities and interactions with healthcare staff. This would help readers better understand the connection between this category and your findings.

• Results: It would be helpful to indicate the sample size (“n”) to clarify what is meant by terms such as “many participants,” “most participants,” and “almost all study participants.” It might also be useful to identify which reasons were most frequently reported overall and within each SEM category. While not all factors may be actionable, highlighting the most commonly cited reasons could help identify priority areas for improving vaccine acceptance, for example, noting if concerns about vaccine safety were among the most frequently mentioned barriers, or if healthcare provider recommendations were among the strongest facilitators (and how many of the 30 participants reported each).

• Discussion (p.25): “Consistent with established frameworks of vaccination…” > It would be interesting to clarify in the methods section why Bronfenbrenner’s socio-ecological model was chosen over the more vaccination-specific WHO BeSD framework. Since the findings appear to align with all BeSD domains, I would recommend briefly stating these domains (with their names and short descriptions) and reflecting on how the two frameworks (Bronfenbrenner’s and BeSD) relate to each other and to your findings.

• Discussion (p.25): “…concerns about vaccine safety was the most common reason for not getting vaccinated.” > This finding is well documented in the existing literature. To strengthen the discussion, you could consider adding references to other studies reporting similar results, both general studies on vaccine hesitancy and, ideally, those specifically focused on immigrant populations.

• Discussion re: limitations (p.29-30): I would suggest adding information about potential bias introduced by excluding individuals who were unable to pay for transportation or participate via videoconference (as mentioned in the methods section).

• Conclusions (p.30): “Further studies among immigrants of different age groups and from other countries are needed for recommended vaccinations with low uptake rates as vaccination decision-making is context- and vaccine-specific.” > I agree with this statement. However, given the focus of your study, I would suggest being more specific by recommending further research on Korean immigrants across different age groups and in cities beyond Montreal and Toronto, as well as on the most commonly reported reasons for low vaccine uptake and potential strategies to address them based on your findings.

**Reviewer #2:** The study is well-designed and the methodology is sound. The discussion is excellent.

Some aspects can be revised to improve the manuscript further. First, tighten the introduction, the paragraph regarding the background of Korean immigration can be reduced into a sentence. Consider adding a comparison of Korean senior vaccination rates versus the national average and other minorities. In my understanding, Koreans have a higher prevalence of vaccination compared to other visible minorities other than others in the East Asian communities but my understanding can be outdated or only applied to COVID-19 vaccines. Table 1 can be made prettier and more condensed. Excellent work relevant and timely for Korean Canadians.

**Do you want your identity to be public for this peer review?** For information about this choice, including consent withdrawal, please see our Privacy Policy

Reviewer #1: No

Reviewer #2: **Yes: ** Yunju Song

---

## [Editor Report · Decision Letter 1]

9 Nov 2025

Vaccination experiences and decision-making in older adult Korean immigrants living in Canada: A qualitative descriptive study

PGPH-D-25-01841R1

Dear Ji Yoon Kim,

We are pleased to inform you that your manuscript 'Vaccination experiences and decision-making in older adult Korean immigrants living in Canada: A qualitative descriptive study' has been provisionally accepted for publication in PLOS Global Public Health.

Best regards,

Baldeep Kaur Dhaliwal, PhD

Academic Editor